# A Fine-Tuning of the Plant Hormones, Polyamines and Osmolytes by Ectomycorrhizal Fungi Enhances Drought Tolerance in Pedunculate Oak

**DOI:** 10.3390/ijms24087510

**Published:** 2023-04-19

**Authors:** Marko Kebert, Saša Kostić, Srđan Stojnić, Eleonora Čapelja, Anđelina Gavranović Markić, Martina Zorić, Lazar Kesić, Victor Flors

**Affiliations:** 1Institute of Lowland Forestry and Environment, University of Novi Sad, Antona Čehova 13d, 21000 Novi Sad, Serbia; 2Faculty of Science, University of Novi Sad, Trg Dositeja Obradovića 3, 21000 Novi Sad, Serbia; 3Division for Silviculture, Croatian Forest Research Institute, Cvjetno Naselje 41, 10450 Jastrebarsko, Croatia; 4Plant Immunity and Biochemistry Group, Department of Biology, Biochemistry, and Natural Sciences, Jaume I University, 12071 Castellón de la Plana, Spain

**Keywords:** drought, pedunculate oak, ectomycorrhizal fungi, polyamines, plant hormones, osmolytes

## Abstract

The drought sensitivity of the pedunculate oak (*Quercus robur* L.) poses a threat to its survival in light of climate change. Mycorrhizal fungi, which orchestrate biogeochemical cycles and particularly have an impact on the plant’s defense mechanisms and metabolism of carbon, nitrogen, and phosphorus, are among the microbes that play a significant role in the mitigation of the effects of climate change on trees. The study’s main objectives were to determine whether ectomycorrhizal (ECM) fungi alleviate the effects of drought stress in pedunculate oak and to investigate their priming properties. The effects of two levels of drought (mild and severe, corresponding to 60% and 30% of field capacity, respectively) on the biochemical response of pedunculate oak were examined in the presence and absence of ectomycorrhizal fungi. To examine whether the ectomycorrhizal fungi modulate the drought tolerance of pedunculate oak, levels of plant hormones and polyamines were quantified using UPLC-TQS and HPLC-FD techniques in addition to gas exchange measurements and the main osmolyte amounts (glycine betaine-GB and proline-PRO) which were determined spectrophotometrically. Droughts increased the accumulation of osmolytes, such as proline and glycine betaine, as well as higher polyamines (spermidine and spermine) levels and decreased putrescine levels in both, mycorrhized and non-mycorrhized oak seedlings. In addition to amplifying the response of oak to severe drought in terms of inducible proline and abscisic acid (ABA) levels, inoculation with ECM fungi significantly increased the constitutive levels of glycine betaine, spermine, and spermidine regardless of drought stress. This study found that compared to non-mycorrhized oak seedlings, unstressed ECM-inoculated oak seedlings had higher levels of salicylic (SA) and abscisic acid (ABA) but not jasmonic acid (JA), indicating a priming mechanism of ECM is conveyed via these plant hormones. According to a PCA analysis, the effect of drought was linked to the variability of parameters along the PC1 axe, such as osmolytes PRO, GB, polyamines, and plant hormones such as JA, JA-Ile, SAG, and SGE, whereas mycorrhization was more closely associated with the parameters gathered around the PC2 axe (SA, ODPA, ABA, and E). These findings highlight the beneficial function of the ectomycorrhizal fungi, in particular *Scleroderma citrinum*, in reducing the effects of drought stress in pedunculate oak.

## 1. Introduction

Climate change has emerged as the most serious threat and challenge to society and the survival of natural ecosystems at the turn of the century. Climate change, which is characterized by an increase in warm and a decrease in cold temperature extremes, as well as shifts in precipitation patterns that may result in prolonged summer drought periods, will have an impact not only on the health and vitality of forests but also on the xeric distributional limits of various tree species by shifting forest distribution patterns to the north-east [1,2,3,4]. According to one of the various climate change scenarios for South and Eastern Europe, droughts will unavoidably become more severe in these areas of Europe (including Serbia), causing significant changes in bioclimatic niches for most tree species [5]. Despite its economic and ecological importance [6,7], the pedunculate oak (*Quercus robur* L.) is considered the species most threatened by climate change in the region [1]. Indeed, the increased drought sensitivity of the pedunculate oak has been demonstrated not only by a stable carbon isotope ratio (δ^13^C) and radial growth approach [2] but also by a biochemical approach in comparison to other oak species (e.g., Turkey oak) [3]. The latter study focuses on the species-specific mechanisms of the pedunculate oak to cope with imposed drought, but it ignores the enormous role of microbes in drought stress mitigation.

Furthermore, climate change alters soil microbiome signatures, and the diversity and activities of microbial communities have a significant impact on plant health and resistance or tolerance to stressors, particularly drought [4,8]. There is extensive above-belowground chemical communication between plants and microbes because plants actively shape the microbial community structure through exudate signaling modulation, whereas soil microbes prime and boost plant immunity as well as reprogram and orchestrate entire plant responses to various stressors including drought [9,10,11,12]. A plant’s ability to communicate with the rhizobiome-root microbiome, which functions as an interface between the plant and the soil environment, increases during a drought due to an increase in exudate biosynthesis [13] and facilitates the exchange of nutrients and metabolites and increases overall plant-microbe interplay in order to mitigate plant stress [4]. On the other hand, drought simultaneously affects the microbial abundance, structure, and activity [14,15]. Ectomycorrhizal fungi (ECM) are among the most abundant and important microbes that colonize pedunculate oak, with 23 taxa of ECM detected and molecularly identified on fine roots of the pedunculate oak [16]. Recently, ectomycorrhizal fungi have been proven to mitigate heat stress in pedunculate oaks [17], as well as alleviate infection with powdery mildew, a foliar fungal disease caused by *Erysiphe alphitoides* [(Griffon and Maubl.) U. Braun and S. Takam.] and *E. quercicola* (S. Takam and U. Braun) [18]. Although it is well known that ECM fungi significantly increase the plants’ root absorptive area, which supports the oaks’ rhythmic growth [19] (especially during the summer’s drought [20]), the specific molecular mechanisms of how ectomycorrhizal fungi affect the plant’s growth regulators and the oak’s drought tolerance mechanisms remain unknown.

Plants evolved very complex drought tolerance mechanisms to combat the concomitant effects of drought such as osmotic and oxidative stress, which include activation of antioxidant defense machinery and rapid accumulation of osmoprotective compounds [21,22]. Various low molecular weight compounds with polar groups, such as carbohydrates (mannitol, trehalose, and raffinose family oligomers (RFO)), polyols, quaternary ammonium compounds, amino acids (such as proline) or polyamines, could be biosynthesized and accumulated as so-called cytoprotectants during droughts as osmotic adjustments to protect cell contents against abiotic stress and maintain cell vital turgor [21,22]. Proline (PRO), a multifunctional amino acid and powerful antioxidant compound, is one of the most commonly used biomarkers of drought stress [23], but it has recently been discovered that mycorrhizal fungi also influence its accumulation and metabolism [24,25]. An increase of osmolytes such as proline, glycine-betaine, and dimethylsulphoniopropionate (DMSP) (sulfur analog of glycine-betaine) during drought stress in pedunculate oaks has previously been reported [3], but without consideration of the effects of ectomycorrhizal fungi and their influence on oak drought tolerance modulation.

On the other hand, while polyamines are well established as a marker of abiotic and biotic stress in vegetables and crops [26], data on woody plant species are very scarce [17,27]. Polyamines, as ubiquitous aliphatic polycations that regulate growth, transcription, and post-translational processes in plants, are also important antioxidants with a high ROS-scavenging capacity in both free and conjugated forms [28]. Polyamines have a high potential for reducing drought stress in plants, not only because of their antioxidant and osmoprotective properties but also because they could be a source of nitric oxide and ROS in guard cells. Polyamines are also crucial signaling molecules that play an important role in stomatal closure and water loss prevention [29,30]. As a result, many studies have demonstrated that the exogenous application of polyamines alleviated drought stress by modulating other osmolytes via cross-talk [31]. Furthermore, mycorrhizal fungi modulate polyamine and aquaporin metabolism, which act as conduits for water stress signals, and alter their expression patterns during drought stress [32]. Despite the fact that some mechanisms of arbuscular mycorrhiza-induced drought alleviation have been reported [33], data on the effects of ectomycorrhizal fungi on polyamine accumulation in pedunculate oak during drought stress are still lacking.

Phytohormones are known to govern a variety of vital processes in plant cells, including both drought adaptation as well as plant-microbe symbiosis establishment [34,35]. Furthermore, a large body of literature has been devoted over the decades to elucidating the most studied relationship between a stress hormone abscisic acid (ABA) signaling and drought tolerance, which happens via the regulation of stomata closure, root development, and osmotic adjustments [36,37,38,39]. Moreover, mycorrhizal fungi are known to prime and reprogram plant immunity and defense mechanisms by modulating plant hormones such as jasmonic acid (JA) and salicylic acid (JA), as well as to improve tolerance to abiotic and biotic stress [40,41], but these mechanisms in woody plant species are understudied. Furthermore, there is still a knowledge gap regarding the role of other hormones in drought tolerance, as well as the synergism and cross-talk between other hormones and signaling molecules, such as ROS, and plant growth regulators, such as polyamines [39,42].

Therefore, the main aim of this study is to mechanistically elucidate the mitigating potential of ectomycorrhizal fungi to drought stress in pedunculate oak seedlings. To complete the aim, we tested the following hypothesis on whether inoculation with ectomycorrhizal fungi modulates and effects: (i) the photosynthetic parameters (net rate of photosynthesis, stomatal conductance, and transpiration rate), (ii) nitrogen and carbon content, and (iii) amounts of osmoprotectants such as proline and glycine betaine during applied drought stress in pedunculate oaks. Furthermore, we tracked the influence of ectomycorrhizal fungi and drought on levels of other metabolites involved in the regulation of stress responses such as (iv) polyamines (putrescine, spermidine, and spermine) and the main (v) plant hormones involved in stress responses (abscisic acid, jasmonic acid, 12-oxophytodienic acid (OPDA), salicylic acid, and its conjugates).

## 2. Results

### 2.1. Drought Effect on Carbon, Nitrogen, and Leaf-Relative Water Contents in ECM and NM Seedlings

Mild drought (MD) and severe drought (SD) both reduced the leaf relative water content in ectomycorrhizal (ECM) and non-mycorrhizal (NM) seedlings, but this effect was more prominent in NM plants. Relative water content (RWC) was significantly lower in NM plants (58.4% and 55.3% in NM plants imposed to MD and SD respectively vs. 65.5% and 63% under MD and SD in ECM plants). Our findings also indicate that mycorrhization increases RWC since unstressed inoculated oak seedlings showed significantly higher leaf water content (72.5%) than non-mycorrhized control plants (61%) (Figure 1a). Two-way ANOVA has shown that mycorrhization (F 82.23) alone had more prominent effects on RWC than drought (F 21.313) (Appendix A).

Similarly, unstressed oak seedlings inoculated with ectomycorrhizal inoculum had a higher nitrogen leaf content than non-mycorrhized oak seedlings. Drought treatments, both severe and mild, caused a significant decrease in nitrogen levels (by 7.7%) in mycorrhized plants, but in NM plants, a significant increase in N content was observed in SD treatment (by 16%) compared to untreated controls. The nitrogen leaf content in mycorrhized oak seedlings ranged from 24.9 to 27.3 mg g^−1^ DW while in non-mycorrhized plants ranged between 25.2–39 mg g^−1^ DW (Figure 1b). Likewise, the effect of treatment interaction (ECM x Drought) had a more pronounced effect on N content modulation in treated oak seedlings (F 13.035) than in each treatment individually (Appendix A).

Within ECM seedlings, mild drought (MD) had no effect on carbon content, whereas severe drought (SD) treatment caused a 1,1% decrease in carbon content. The carbon content in the mycorrhized control plants was significantly higher (438 mg C g^−1^ DW) than in the non-mycorrhized control plants (422 mg C g^−1^ DW) (Figure 1c). Contrary to the N content, a higher sample variation was not provided by the synergistic effects of ECM and drought (Appendix A).

Under progressive drought stress, both ECM and NM seedlings showed a decreasing trend in gas exchange parameters (Figure 2). As drought stress increased, A, E, gs, and WUE values decreased in both treatments, reaching their lowest levels in seedlings subjected to severe drought. When compared to the control group of plants, a sharp decline of A, E, and gs was evidenced in ECM seedlings exposed to mild drought, while WUE decreased gradually through observed drought treatments. In contrast, this trend was less pronounced between ECM plants subjected to MD and SD, although the differences in gs values were statistically significant between different drought treatments (i.e., no statistical differences were observed in MD and SD plants in terms of A and E). In the NM seedlings, mild drought had no effect on E and gs, while significant changes occurred only during severe drought. Nevertheless, compared to NM plants, the ECM seedlings have been characterized with higher mean values of all parameters when observed across individual treatments. All measured leaf gas-exchange parameters statistically varied in response to both the analyzed factors (ECM and drought) as well as their interactions, as shown by the results of the ANOVA.

### 2.2. Drought Effect on Osmolyte Levels in the Presence and Absence of ECM Fungi

Both mild and severe drought treatments significantly increased the free proline content in the ECM and NM seedlings. In the mycorrhized plants, severe drought significantly increased the free proline induction compared to mild drought, with severe drought causing a 2.5-fold increase and mild drought causing a 1.66-fold increase compared to the untreated control. Both severe and mild drought induced proline induction in non-mycorrhized oak seedlings compared to the unstressed control oak seedlings, but there were no significant differences in the free proline content between the two levels of imposed drought (Figure 3a). Two-way ANOVA showed that drought (F value 287.411) had a stronger effect than mycorrhization (F value 11.207) on the foliar-free proline (Pro) content, while their interaction provided a moderate effect on the proline content (F89.113) (Appendix A).

Water scarcity increased the glycine-betaine content in both the ECM and NM plants, with severe drought having a significantly stronger effect on the NM oak seedlings. Mycorrhization had a significant impact on the glycine-betaine levels, so the foliar levels of the GB in the mycorrhized oak seedlings were approximately 24.2 μmol g^−1^ DW in comparison to 22.4 μmol g^−1^ DW in the NM control. (Figure 3b). In terms of the GB response, drought (F 15.115) had a stronger impact on oak seedlings than mycorrhization (F 0.066), which was not statistically significant at the *p* 0.05 level (Appendix A).

In contrast to the free proline and glycine betaine, the putrescine content decreased in both non-mycorrhized and mycorrhized plants during severe drought. Constitutive foliar putrescine levels of the simplest polyamine, putrescine, were significantly higher in regularly watered oak seedlings inoculated ectomycorrhizal fungi (approximately 275.9 nmol g^−1^ DW) compared to non-mycorrhized unstressed plants (about 166.7 nmol g^−1^ DW) (Figure 3c). Although unstressed plants inoculated with the ectomycorrhizal fungi had significantly higher detectable levels of higher polyamines, spermidine, and spermine than the non-inoculated controls, drought showed a more dramatic increase in both higher polyamines under severe stress in the non-mycorrhized plants (Figure 3d,e). Spermidine was found to be the most abundant polyamine in the oak leaves, followed by putrescine and spermine. The spermidine and spermine levels in non-inoculated plants ranged from 240 to 424.2 nmol g^−1^ DW for spermidine and 88.9 to 222.4 nmol g^−1^ DW for spermine, while these osmolytes ranged from 271.9 to 300.8 nmol g^−1^ DW for spermidine and 111.5 to 169.7 nmol g^−1^ DW for spermine in mycorrhized plants. In ECM seedlings, spermine showed a more prominent increase induced by drought, whereas spermidine increased only slightly. Furthermore, unlike spermine, spermidine did not show any significant differences in terms of accumulation between the two levels of drought (severe and mild) (Figure 3d,e). According to two-way ANOVA, overall, drought has a stronger impact on all individual polyamines than mycorrhization (Appendix A).

Interestingly, free salicylic acid exhibited ambiguous drought dependency trends in regard to mycorrhization. Therefore, under severe drought stress, non-mycorrhized plants had a dramatic increase in SA levels (about 1.78 folds), whereas, under severe and mild drought, the mycorrhized oak seedlings had a significant decrease in SA amounts compared to non-stressed controls. The constitutive levels of SA in regularly watered oak seedlings were notably higher (by 2.28 folds) compared to the unstressed seedlings previously primed with ectomycorrhizal inoculum. The SA levels in the non-mycorrhized oak seedlings ranged from 4135 to 7460 ng SA g^−1^ DW, while the levels in the ECM-inoculated oak seedlings ranged from 5501 to 9669 ng g^−1^ DW (Figure 4a). Following the ANOVA test, both examined effects and their interaction had a statistically significant impact on the SA content of the examined oak seedlings (Appendix A). When the oak seedlings were subjected to both mild and severe drought, the jasmonic acid exhibited similarly increasing patterns compared to the unstressed corresponding controls, regardless of mycorrhization. There were no significant differences in the JA content between the unstressed mycorrhized and the non-mycorrhized controls (Figure 4b). On the other hand, drought significantly influenced the jasmonic response in the analyzed samples (F 57.837) (Appendix A). In contrast, the abscisic acid levels were higher in the control seedlings inoculated with the ectomycorrhizal fungi than in the non-mycorrhized, unstressed, oak seedlings. The response of ABA to severe drought in the mycorrhized plants was significantly stronger than in the non-mycorrhized oak seedlings, with the ECM plants accumulating approximately five-fold more ABA compared to the non-mycorrhized oak plants. Following a two-way ANOVA F-test, it was determined that mycorrhization, drought, and their interaction had a significant impact on the variation in ABA response. Droughts had less of an impact on the SA and ABA response than mycorrhization (Appendix A). Unlike in ECM-inoculated plants, the severity of drought had no effect on the ABA accumulation in non-mycorrhized seedlings (Figure 4c). In contrast to ABA, whose constitutive amounts were found to be more abundant in ectomycorrhizal-inoculated, non-stressed, seedlings compared to the non-inoculated regularly watered controls, the 12-oxo-phytodienoic acid (OPDA) constitutive amounts were found to be more than double in the non-mycorrhized drought unstressed plants. Therefore, OPDA showed different drought-induced trends, depending on mycorrhization, within non-mycorrhized plants showing declining OPDA trends, while the ECM-inoculated plants showed ascending ODPA trends (Figure 4d). The OPDA responses to drought, mycorrhization, and their interaction were almost identical (F values 67.431, 73.792, and 52.268, respectively) (Appendix A). Both conjugated forms of salicylic acid, salicylic acid-2-O-β-d-glucose (SAG), and salicylic acid-glucose ester (SGE) exhibited similar drought inducible patterns, whereas the response to severe drought was stronger in non-mycorrhized plants and indeed stronger than to mild drought. SGE levels of SA conjugates (ranging from 2735 to 5053 ng SGE g^−1^ DW) were significantly higher than levels of SAG conjugates (ranging from 72 to 192 ng SAG g^−1^ DW) (Figure 4e,f). A conjugate of jasmonic acid, jasmonate-isoleucine (JA-Ile) showed drought-responsive patterns very similar to the jasmonic acid with a slight difference, its response was strongly induced by severe drought in the non-mycorrhized plants than in the ECM-inoculated, although its constitutive levels were higher in the ECM-primed plants. The jasmonic acid was found in higher concentrations in a conjugated form of jasmonate-isoleucine than in a free form in the oak seedlings (Figure 4g).

The analyzed metabolites showed various correlation patterns. Individual polyamines exhibited different correlation patterns, so putrescine differed drastically from SPM and SPD (Figure 5). Osmolytes (GB and PRO), higher polyamines (SPD and SPM) and plant hormones J, ODPA, and certain plant hormone derivatives (SAG, SGE, and JA-Ile) showed a mostly negative correlation with the gas exchange parameters (A, E, and gs) and RWC, nitrogen, and carbon content. On the contrary, the polyamine putrescine, plant hormones, SA, and ABA exhibited a mostly positive correlation with the gas exchange parameters, C and N content, and RWC. Unlike the PUT, higher polyamines (SPD and SPM) exhibited a mostly positive correlation with both osmolytes PRO and GB. ABA had a strong positive correlation with PRO, while JA strongly stimulates GB accumulation. The trade-off mechanism between JA and SA is most evident in its correlation with gas exchange parameters (A, E, and gs) and PUT and PRO. JA, ODPA SAG, and SGE showed similar positive correlation patterns with higher polyamines (SPD and SPM), but a strong negative correlation with PUT. JA and JA-Ile showed comparable correlation patterns, while SA and its derivatives (SAG and SGE) showed antagonistic behavior (Figure 5).

The resulting effects of mycorrhization and drought on the biochemical and physiological parameters of pedunculate oak seedlings were estiSupmated using a principal component analysis (PCA). The first two principal components (PCs) of the PCA explain 62.17% of the total variance (Figure 6); whereas PC1 accounts for 43.12% of the total sample variance, and PC2 accounts for 19.05%. The effects of drought and ECM-inoculation effectively separated our samples by PCA in accordance with the parameters examined. By comparing the distance between the non-mycorrhized and mycorrhized controls, we noticed that the mycorrhization factor separated samples along the PC2 axes, which correspond to the SA, ABA, OPDA, and transpiration rate (E) parameters. On the other hand, the drought effect was displayed across PC1 that was defined by the contents of osmolytes (PRO and GB), polyamines (SPD and SPM), and plant hormones such as JA and its conjugate JA-Ile, as well as SGE and SAG. Furthermore, net photosynthesis, stomatal conductance, and leaf water content were more variable under the drought effect and corresponded to PC1. The effects of mild and severe droughts were successfully differentiated within both mycorrhized and non-mycorrhized oak seedlings, mostly along PC1.

## 3. Discussion

The nitrogen levels in the unstressed and regularly watered ECM-inoculated oak seedlings were higher than in non-inoculated seedlings, confirming that the main benefits of this fungi-plant beneficial partnership are not only amelioration [43], but also improved the plant’s nitrogen uptake [44]. Droughts induced a decline in the oak nitrogen content in both the non-mycorrhized and mycorrhized seedlings, which is in accordance with the previous findings [45,46].

The RWC and gas exchange parameters of leaves were frequently used as important indicators of the drought-stress effect on plants [47,48,49]. Accordingly, our results showed that ECM plants displayed greater values of RWC compared to NM plants, indicating that inoculated seedlings were able to maintain a higher water balance inside the leaves than non-inoculated ones. In addition, we found that progressive drought stress greatly reduced the RWC in both the ECM and NM seedlings, although a different pattern of RWC decline was observed in the two treatments. Specifically, it was discovered that the differences between the SD and MD were not statistically different, despite the fact that, in the case of the mycorrhized plants, a sharp decline in RWC was recorded in the MD treatment compared to the control plants. On the other side, a continuous decline of RWC was recorded in the NM seedlings across the treatments. According to Ritchie et al. [50], smaller decreases in the leaf RWC indicate their greater water-retention capacity, as well as a greater adaptability of the leaves to resist drought-induced stress.

The lower dehydration status of the ECM seedlings, measured by RWC, was also confirmed by the gas exchange parameters, indicating a greater photosynthetic capacity (i.e., higher values of A, E, and gs) of the ECM plants under moderate to severe drought stress. Higher C and N concentrations in the ECM seedlings, compared to the NM, suggest that mycorrhizal symbiosis can provide nutritional benefits to the host plants, which can enhance carbon assimilation and N accumulation in the leaves [51]. Additionally, we discovered that the levels of N and C in the leaves positively correlated with the leaf water content and CO_2_ assimilation rate, supporting the conclusion reached by Heinonsalo et al. [52] that mycorrhizal symbiosis has a major impact on the plant’s water status and N content, which in turn affects C allocation and photosynthesis.

Although some examples suggest that this may not always be the case, previous studies have shown that ectomycorrhizal fungi may be essential in reducing biotic and abiotic stresses on the plant’s physiological processes [52]. According to Heinonsalo et al. [52], *Pinus sylvestris* mycorrhizal and non-mycorrhizal seedlings did not significantly differ in any of the investigated photosynthesis-related parameters. However, there are numerous other instances that demonstrate how ectomycorrhizal fungi have a positive impact on the plants’ physiological responses. For example, Wright et al. [53] evidenced that *Betula pendula* plants colonized with the ectomycorrhizal fungus *Paxillus involutus* (Batsch) Fr. exhibited higher rates of photosynthesis after six months of growth. Makita et al. [51] reported a higher leaf photosynthetic rate and nitrogen concentration in *Quercus serrata* seedlings colonized by ectomycorrhizal fungi. Finally, Kebert et al. [3,18] demonstrated a positive effect of ectomycorrhizal on gas exchange in *Quercus robur* seedlings subjected to heat stress and powdery mildew infestation, respectively.

Drought affects and modulates the dynamics of nitrogen or sulfur-containing compounds in oaks, such as amino acids or polyamines [3,54,55], and recent studies confirmed that the osmolyte response and involvement of different osmotically active, nitrogen-containing, metabolites have to drought in different oaks are species and organ-specific, as well as dependent on soil type, drought intensity, and the duration of exposure [3,49,54,55]. The results of this experiment agree with the classical drought-triggered phenomena of increased the accumulation of osmotically-active compounds such as proline, glycine betaine, and main polyamines in most plant species, including oaks, as a well-defined plant response and strategy to compensate for water scarcity [56,57,58]. Proline’s importance as a multifunctional amino acid is well documented and described in the literature, and its plant drought-mitigation is linked to its antioxidant, redox balancing, and osmoprotective properties, as well as its chaperon, signaling, and gene expression modulating role [23,59,60]. Many factors influence the levels of this important drought-mitigating amino acid in plants, including biosynthesis and catabolism, as well as transport and turnover within the plant [61]. There is evidence that drought modifies the proline’s metabolism by upregulating the proline biosynthetic genes *P5CS1* and downregulating the proline catabolic genes *ProDH* [60,62]. In this study, drought induced increasing patterns of foliar proline in both the mycorrhized and non-mycorrhized oak seedlings, whereas in later, the non-stressed controls contained higher constitutive proline levels, whereas severe drought-inducible proline levels were higher in the ECM-inoculated oak seedlings. This data on the enforced proline response to drought in the mycorrhized plants has also been reported in the trifoliate orange tree [63], olive trees [64], and different potato cultivars [65], but was attributed to arbuscular mycorrhizal fungi. Our findings support a strong synergistic interaction between the ectomycorrhizal fungi and proline metabolism in oaks, which is consistent with previous research [66,67]. A strong positive correlation between ABA and proline has also been previously reported [68].

Droughts (both levels, mild and severe) triggered the accumulation of the main quaternary ammonium compound, glycine betaine, in the leaves of the pedunculate oak seedlings. These findings are consistent with the previous findings in which the drought tolerance of many plant species increased due to GB accumulation [69,70], including in *Q. robur* [3]. The mechanism of GB action is not as a direct ROS scavenger, but rather as an indirect activator of various antioxidant enzymes such as peroxidases, catalase, and SOD [69,71]. Because of its positive charge, GB stabilizes and safeguards proteins involved in photosynthesis, such as the oxygen-evolving PSII complex, during drought treatments and shields them from oxidative damage [72]. Unlike proline, the GB constitutive levels were higher in ECM-inoculated, unstressed, plants, but the highest levels of accumulated drought-inducible GB were detected in the non-inoculated oak plants exposed to severe drought. The ECM-inoculated regularly watered controls had a significantly higher GB content than the non-mycorrhized oak seedlings controls, which agrees with the previously reported data of the increasing glycine betaine patterns observed in white seedless grapes inoculated with different arbuscular mycorrhizal fungi (AMF) species (*Funneliformis mosseae*, *Rhizoglomus irregularis*) subjected to drought stress [73].

Polyamines, as ubiquitous low molecular weight polycations with numerous regulative roles in plant cells, play a pivotal role as mediators of abiotic and biotic stress tolerance in plants, as well as conductors of signaling in plant-fungi communication [27,74,75,76]. The roles and mechanism of polyamines alleviation of drought [77,78,79] and heat stress [17], or a combination of drought and heat stress [31] has been previously reported. To the best of the author’s knowledge, this is the first study examining the effects of mycorrhization and drought on pedunculate oaks. Diamine putrescine, the simplest polyamine, was significantly more abundant in the ECM-inoculated, well-watered, seedlings compared to the non-mycorrhized controls, and these findings are consistent with the increased putrescine levels and the upregulated expression of biosynthetic genes for ornithine decarboxylase *PtODC* in the orange tree (*Poncirus trifoliata* L.) after AMF inoculation [80,81]. Severe drought (SD) stress reduced the putrescine accumulation significantly in both the ECM and NM oak seedlings, which may be attributed to the fact that putrescine is a precursor for both prolines as well as biosynthesis of higher polyamines, spermine, and spermidine [80,82,83].

Higher levels of spermine and spermidine in the ECM-inoculated seedlings compared to the non-mycorrhized seedlings could be linked to the role of polyamines in root architecture, but also in early mycorrhiza-host recognition and plant-fungi dialogue, as well as mycorrhiza development via mycelial growth stimulation [81,84]. Higher polyamines, spermine, and spermidine exhibited similar drought-induced increasing trends that differed from the putrescine behavior patterns in this experiment. Polyamines alleviate drought in plants by modulating the leaf water status and photosynthesis, as well as their polycationic nature allows them to stabilize their thylakoid membranes through chaperone activity [85]. During polyamine catabolism, large amounts of signaling molecules such as NO and H_2_O_2_ are produced, which are pivotal in stomatal movement and closure, preventing water loss during drought and influencing redox balance and antioxidant system improvement [86,87,88].

A large body of literature confirmed that the dialogue between fungi and plants is communicated via plant hormones and that arbuscular mycorrhiza fungi are capable of reprogramming the entire primary and secondary plant metabolism [89,90]. Despite the fact that there has been a lot of research on AMF, only a few have been dedicated to ectomycorrhizal fungi (ECM) and their reprogramming of oak metabolisms, particularly at the lipidomic level [91,92], but this is one of the pioneering research focused on how ECM fungi reprogram oak seedlings at the plant hormone level and affect their drought tolerance. This study discovered that unstressed, ECM-inoculated oak seedlings had higher levels of SA and ABA, but not JA acid, than non-mycorrhized oak seedlings, which is consistent with previous findings that outlined that salicylic acid (SA) orchestrates defense mechanisms triggered against biotrophic pathogens, which are more similar to ectomycorrhizal fungi by their lifestyle than necrotrophs against which the plant activates the JA signaling pathways [93,94]. Droughts silenced the SA response in oak seedlings that were colonized with the ECM inoculum. This seems to be the result of a storage mechanism in which free SA is converted into glucoside’s inactive form. On the other hand, with non-mycorrhized oak seedlings, severe drought induced levels of free SA. The underlying mechanisms of how SA affects drought stress were previously described [42,95,96]. Furthermore, it was demonstrated that arbuscular mycorrhiza fungi, in synergy with SA, modify the hydraulic responses in maize during drought by modulating the aquaporin metabolism [97]. SA can be glycosylated either on hydroxyl or a carboxylic group by activities of different UDP-glucosyltransferases, resulting in the formation of different conjugates salicylate-2-O-β-D-glucoside (SAG) or salicylate glucose ester (SGE), respectively [98]. In our study, both conjugates showed similar drought-induced increasing patterns, but a more prominent response to severe drought was detected in the non-mycorrhized oak seedlings, in terms of the accumulation of these conjugates. While the SAG form serves as long-term storage that is transported into the vacuole, the SGE form is more easily hydrolyzed and more accessible to release active hormone SA [99]. Increased levels of SA under water deficit were also reported in *Phillyrea angustifolia* plants (five-fold) [100] as well as in barley (two-fold) [34,101]. Mycorrhization did not have any effect on JA accumulation, but the levels of JA acid increased drastically under the severe drought stress in both mycorrhized and non-mycorrhized oak seedlings. JA, a lipid-derived phytohormone best known for its role in wounding, is also involved in the alleviation of abiotic stresses. A dramatic increase in JA triggered by the severe drought in this study is consistent with previous research on *Q. robur* in which seed desiccation strongly induced JA pathways [102]. A mitigation effect of JA to drought has been demonstrated when JA was applied exogenously to maize, barley, rice, soybean, and banana [103]. The mechanism of JA is related to the activation of antioxidant enzymes such as peroxidases, catalase, and SOD, as well as the upregulation of proline [104].

Undoubtedly, abscisic acid, a zeaxanthin-derived plant hormone, is the most referenced plant hormone whose increased accumulation is linked to drought tolerance in many plant species due to ABA’s ability to regulate many stress-related genes that affect drought tolerance in plants, as well as ABA’s capability to affect guard cell turgor and induces stomata closure via complex signal transduction [34,37]. Drought’s induced increment of ABA levels from this study is in accordance with previously reported increases in ABA levels in *Q. cerris* and *Q. robur* exposed to drought [3], while a comprehensive transcriptomic study on *Q. robur* and *Q. pubescens* revealed that drought altered expression patterns of three key genes involved in ABA signaling pathways, including PP2C27 (Protein phosphatase 2C 27), RD22 (Response to desiccation 22), and STP13 (Sugar transport protein 13) [105]. The ABA response to drought was more pronounced in oak seedlings inoculated with ectomycorrhizal fungi, which is consistent with previous findings that the AMF mycorrhizal symbiosis influences the ABA metabolism in *Vitis vinifera* L. under different irrigation regimes [106] as well as an osmolyte and antioxidant accumulation in *Nicotiana tabacum* L. [107]. In our study, the ABA amounts were increased in oak seedlings inoculated with ectomycorrhizal fungi *Scleroderma citrinum* regardless of imposed drought stress. Similarly, ectomycorrhizal fungi *Laccaria bicolor* improved drought tolerance in poplars through an increased ABA root accumulation and a downregulation of aquaporins (*PtPIP1;4*, *PtPIP2;3*, and *PtPIP2;10*) expression [108]. The importance of ABA accumulation and the upregulation of carotenoid pathways in the establishment of symbiotic relationships with ectomycorrhizal fungi *Pisolithus microcarpus* has also been found on *Eucalyptus grandis* root tips [109]. Interestingly, it was discovered that some ectomycorrhizal fungi (*Sistotrema sp.*) can even produce ABA during symbiosis establishment with *Quercus virginiana* seedlings [110].

One of the most prominent members of oxylipins, oxygenated derivates of fatty acids (FAs), is 12-oxo-phytodienoic acid (OPDA), which is best known as a major jasmonate precursor and regulator of jasmonate-responsive genes that has an important role in plant defense and stress acclimation [111]. The ODPA levels in mycorrhized plants were more responsive to drought than in non-inoculated seedlings, where drought caused a significant decline. This decline in ODPA levels in non-mycorrhized plants could be attributed to the fact that OPDA was used as a precursor that undergoes the process of β-oxidation to form jasmonate (JA), who’s content simultaneously increased significantly under severe drought in non-mycorrhized seedlings. Drought-tolerant varieties of *Cicer arietinum* L. showed an increased expression in oxylipin genes, increased root accumulation of ODPA, and concurrent accumulation of its derivatives, JA and JA-Ile [112]. Mycorrhization induced a decrease in the OPDA levels in oak seedlings, which is related to the fact that AMF alleviates oxidative stress in plants, thus lessening lipid peroxidation in plants [113]. However, the exact role of oxylipins and their biosynthetic genes that code different lipoxygenases (LOXs) in the establishment of host mycorrhiza symbiosis is still not fully understood. For this reason, some studies have confirmed that only the 9-LOX product was elevated during symbiosis formation and not the 13-LOX product [114], while other studies have suggested that the 13-LOX product was elevated [115], and still other studies have failed to detect any significant differences between mycorrhized and non-mycorrhized [116].

## 4. Materials and Methods

### 4.1. Experimental Design

The experiment was designed in a greenhouse under semi-controlled conditions with drought as the main factor and ectomycorrhizal fungi as a secondary factor, with the goal of estimating how these factors affect gas exchange, polyamine, and osmoprotectants accumulation as well as the plant hormones levels in oak seedlings.

Seedlings of the pedunculate oak (*Q. robur* L.) were germinated from acorns and a soil mixture was prepared as it was thoroughly described in [18]. The forest soil used for the experiment was collected at the locality Višnjićevo, which is situated within the largest pedunculate oak complex in Serbia. The coordinates of this locality, as well as the basic soil properties, are given by Vasić et al. [117].

After six weeks, representative oak seedlings were inoculated with ECTOVIT (Symbiom, s.r.o., Lanškroun, Czech Republic), commercial inoculum containing mycelia, and spores from six species of ectomycorrhizal fungi [18]. Following the 24-week period, after the ectomycorrhizal was confirmed to be developed, DNA was extracted from the ectomycorrhizal root tips using a Qiagen DNeasy plant mini kit (Qiagen, Hilden, Germany) for the molecular identification of the fungi, which was accomplished through PCR amplification and sequencing of the nuclear ribosomal DNA’s internal transcribed spacer (ITS) region [18]. Macrogen Europe performed the sequencing, and the sequence was deposited in GenBank (accession number: ON239799). *Scleroderma citrinum* was the only ectomycorrhizal species identified [18]. Twenty-four weeks after inoculation fully developed plants of both groups, the inoculated (ECM) and non-inoculated (NM) oak seedlings were exposed to the two levels of drought intensity. Drought stress was imposed on both groups of plants. The non-mycorrhized (NM) and the oak seedlings were inoculated with the ectomycorrhizal inoculum (ECM) by withholding watering until the soil water content fell to predetermined levels, 60% and 30% of the total water field capacity (FC), for mild drought (MD) and severe drought (SD), respectively. Each treatment consisted of 10 pots with one oak seedling, for a total of 60 seedlings. During the experiment, the seedlings were cultivated under ambient light conditions, without any additional illumination. The temperature and humidity were controlled in the ranges close to the ambient conditions outside the greenhouse. The air temperature in the greenhouse was maintained between 20 and 30 °C throughout the experiment, while the relative humidity ranged between 45–75%. Pots were weighed daily and watered as required and calculated to keep the soil water content close to the target values. The duration of drought treatments (ECM-SD, ECM-MD, NM-SD, and NM-MD) was 10 days (6–15th September 2020). Both the mycorrhized and non-mycorrhized control oak seedlings (NM-C and ECM-C) were regularly watered for 10 days to reach and maintain full soil water capacity (between 90 and 100%).

### 4.2. Physiological Measurements

Physiological measurements were taken on healthy, fully formed, leaves in the top third of the seedlings. The measurements were taken on plants from 9:00 a.m. to 11:00 a.m. The highly accurate differential gas analyzer CIRAS-3 (PP Systems International, Amesbury, MA, USA) was used to measure the assimilation rate (A, μmol CO_2_ m^−2^ s^−1^), stomatal conductance (gs, mol m^−2^ s^−1^), transpiration rate (E, mmol H_2_O m^−2^ s^−1^), and water use efficiency (WUE = A E^−1^, mol mmol^−1^). The saturating photosynthetic active radiation (PAR) in the leaf chamber was set to 1000 μmol m^−2^ s^−1^, while the CO_2_ concentration, air temperature, and relative humidity were taken in the ambient. In each treatment, 10 seedlings were used for measurements, taking five seedlings from each treatment in two cycles.

### 4.3. Quantification of Osmolytes’ Accumulation

Chemical analyses were conducted in five biological replicates, meaning samples were taken from five seedlings within each treatment. Each sample contained five fully developed upper leaves from the oak seedlings. Samples were ground and powdered in liquid nitrogen, then weighted and used for separate extraction methods.

Polyamines (putrescine (Put), spermidine (Spd), and spermine (Spm)) were extracted from freeze-dried oak leaves (approx. 20 mg DW) with 2 mL of perchloric acid (4% *v*/*v*), homogenized in a ball mill, and then centrifuged at 15,000× *g* for 30 min. The polyamines were derivatized using dansyl-chloride, as previously described by Scaramagli et al. [118]. The separation of different polyamine dansyl-derivateives was performed using a reverse phase C18 column (Spherisorb ODS2, 5-μm particle diameter, 4.6 × 250 mm, Waters, Wexford, Ireland) by high-performance liquid chromatography (HPLC), coupled with fluorescent detection (Shimadzu, Kyoto, Japan) by employing the acetonitrile-water gradient as previously described [118].After dyeing with ninhydrin reagent, the free proline was determined spectrophotometrically according to Bates et al. [119] with minor modifications. Around 20 mg of freeze-dried powdered leaf material was homogenized with 1 mL of sulfosalicylic acid (3% *w*/*v*) and centrifuged for 10 min at 4000 rpm. A total of 0.7 mL of the supernatant was mixed with 0.7 mL of acid ninhydrin solution (2.5% ninhydrin in glacial acetic acid-distilled water-85% orthophosphoric acid (6:3:1)) and 0.7 mL of glacial acetic acid and the reaction mixture was boiled at +95 °C for 1 h and then transferred to ice bath. The pinkish proline-ninhydrin complex compound was extracted with 2 mL of toluene with vigorous vortexing and the absorbances were recorded at 520 nm by using a MultiScan GO (ThermoScientific, Bremen, Germany) spectrophotometer.The concentration of glycine-betaine (GB), as the predominant quaternary ammonium compound (QAC), was determined spectrophotometrically using the precipitation method of QAC-periodide complexes in an acid medium given by Grieve and Grattan [120].

### 4.4. Plant Hormones Analysis

The oak leaves were freeze-dried and powdered in liquid nitrogen, and weights of about 30 mg were used for plant hormone extraction. About 30 mg of leaf material was mixed with 1 mL of extraction mixture (30% MeOH in water) and stabilized with 0.01% formic acid containing 5 ppb of the internal standard mixture (abscisic acid-d6 (ABA-d6), salicylic acid-d5 (SA-d_5_), indole acetic acid-d5 (IAA-d5), jasmonic acid-d_5_ (JA-d_5_), and jasmonate isoleucine-d_6_ (JA-Ile-d_6_) in 2 mL Eppendorf tubes. The extraction and homogenization were aided by using a ball mill (Retsch, model MM 500, Haan, Germany) for 30 s on each tube, followed by 30 min on ice and 15 min at 15,000 rpm at 4 °C. The supernatant was collected and filtered through a syringe filter (25 mm, 0.22 μm) into vials, which were then diluted three times before 20 μL was injected into an ultra-performance Waters Acquity liquid chromatography (UPLC™) system (Waters, http://www.waters.com). An orthogonal Z-spray electrospray interface was used to connect the UPLC to a triple quadrupole tandem mass spectrometer (TQS; UPLC; Waters, Milford, MA, USA). At a flow rate of 0.3 mL.min^−1^, a UPLC Kinetex C18 analytical column with a 5 μm particle size, 2.1 100 mm (Phenomenex, Torrance, CA, USA), was used for LC separation [121]. All hormone standard curves were obtained by injecting a mixture of pure compounds at various concentrations (0.5, 1.25, 2.5, 5, and 7 μg L^−1^). Mass Lynx v4.1 (Waters) was used to perform the quantifications, with the internal standards serving as a reference for extraction recovery and the standard curves serving as quantifiers.

### 4.5. Carbon and Nitrogen Elemental Analysis

Small tin capsules containing 20 mg of powdered, freeze-dried, oak leaf samples were burned in an elemental analyzer’s combustion box for about 2 min at 900 °C (model Elementar Vario EL III, Germany). Carbon and nitrogen were converted from samples into CO_2_ and N2, respectively. Analytical columns were used to separate the gases, and variations in the thermal conductivity of formed gasses were used to quantify them relative to the standard. Acetanilide (C 71.09% and N 10.36%) was used as a standard and the percentages of C and N were automatically calculated in samples [122].

### 4.6. Statistical Analysis

The following statistical methods were used: descriptive statistics, two-factorial ANOVA, the t-test, principal component analysis (PCA), and Pearson correlation. Drought and ectomycorrhizal were factors in a two-way ANOVA, and their significance levels and Fisher (F) test results were used to interpret them. A box-plot diagram served as a visual representation of the t-test results. All the statistical data processing was performed using the R programming language (R Core Team). Descriptive statistics, two-way ANOVA, and t-tests were computed using the “rstatix” R package [123] and the other visual representations were created using the “ggplot2” R package [124].

## 5. Conclusions

This study is a pioneering investigation into how ectomycorrhizal fungi affect the biochemical characteristics in drought-exposed pedunculate oaks. We proved that ectomycorrhizal fungi inoculation modifies the drought stress response in pedunculate oaks. Particularly, in the presence of ectomycorrhizal fungi, proline and the ABA response in pedunculate oaks were amplified during severe drought stress. On the other hand, regardless of drought stress, the constitutive levels of all polyamines (Put, Spd, and Spm), both inspected osmolytes (GB and PRO), SA, ABA, and JA-Ile were higher in oak seedlings colonized with ectomycorrhizal inoculum than in the non-mycorrhized controls. By using a PCA analysis, the effects of drought and ECM on the oak’s biochemical response could be distinguished clearly. Mycorrhization, for example, was more closely associated with the parameters gathered around the PC2 axe (SA, ODPA, ABA, and E), whereas the effect of drought was associated with the variability of parameters along the PC1 axe, such as osmolytes PRO, GB, PAs, and plant hormones such as JA, JA-Ile, SAG, and SGE. Mycorrhization showed an alleviating effect to severe drought stress, while the biochemical response to severe drought was gradually amplified in comparison to mild drought more prominent in the non-mycorrhized oak seedlings. These findings demonstrated a significant role for ectomycorrhizal fungi (particularly *Scleroderma citrinum*) in modulating osmolytes (GB and PRO), polyamines, and the priming effect by upregulating specific plant hormones during drought, which all contribute to the increased drought tolerance of pedunculate oak seedlings.

## Figures and Tables

**Figure 1 ijms-24-07510-f001:**
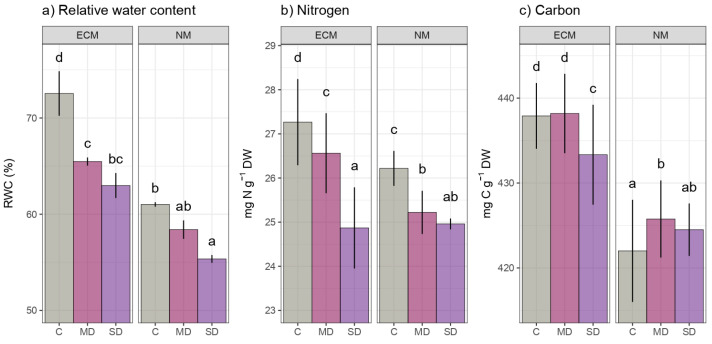
The effects of mycorrhization and/or drought on:(**a**) Relative water content (RWC; %), (**b**) Nitrogen content (N; mg g^−1^), and (**c**) Carbon content (C; mg g^−1^) in leaves of *Q. robur*. Treatments: C-control regularly watered oak seedlings, MD-oak seedlings exposed to the mild drought that corresponds with 60% of soil water field capacity, SD- oak seedlings exposed to the severe drought that corresponds with 30% of soil water field capacity ECM-oak seedlings inoculated with ectomycorrhizal inoculum, and NM-oak seedlings not inoculated with ectomycorrhizal inoculum. Different small letters indicate significant differences across the different treatments; Tukey’s honestly significant difference (HSD) post hoc test (*p* ≤ 0.05). Data represent the mean ± standard deviation (SD).

**Figure 2 ijms-24-07510-f002:**
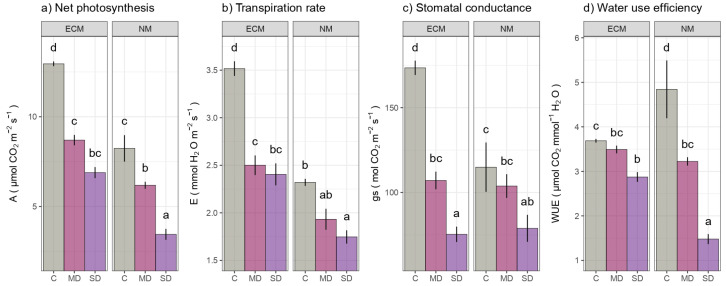
The effects of mycorrhization and/or drought on gas exchange: (**a**) net photosynthetic rate (A; mmol CO_2_ m^−2^ s^−1^), (**b**) Transpiration rate (E; mmol H_2_O m^−2^ s^−1^), (**c**) Stomatal conductance (gs, mmol CO_2_ m^−2^ s^−1^), (**d**) Water use efficiency (WUE; mmol CO_2_ mmol^−1^ H_2_O) of leaves of *Q. robur*. Treatments: C-control regularly watered oak seedlings, MD-oak seedlings exposed to the mild drought that corresponds with 60% of soil water field capacity, SD- oak seedlings exposed to the severe drought that corresponds with 30% of soil water field capacity ECM-oak seedlings inoculated with ectomycorrhizal inoculum, and NM-oak seedlings not inoculated with ectomycorrhizal inoculum. Different small letters indicate significant differences across the different treatments; Tukey’s honestly significant difference (HSD) post hoc test (*p* ≤ 0.05). Data represent the mean ± standard deviation (SD).

**Figure 3 ijms-24-07510-f003:**
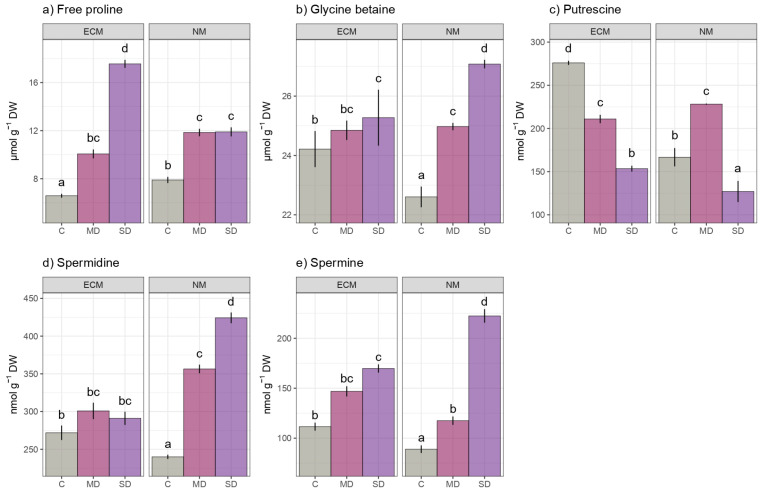
The effects of mycorrhization and/or drought on osmolyte levels: (**a**) Proline (PRO; mmol g^−1^ DW), (**b**) Glycine betaine (GB; mmol g^−1^ DW), (**c**) Putrescine (PUT; nmol g^−1^ DW), (**d**) Spermidine (SPD; nmol g^−1^ DW), (**e**) Spermine (SPM; nmol g^−1^ DW) in leaves of *Q. robur*. Treatments: C-control regularly watered oak seedlings, MD-oak seedlings exposed to the mild drought that corresponds with 60% of soil water field capacity, SD- oak seedlings exposed to the severe drought that corresponds with 30% of soil water field capacity ECM-oak seedlings inoculated with ectomycorrhizal inoculum, and NM-oak seedlings not inoculated with ectomycorrhizal inoculum. Different small letters indicate significant differences across the different treatments; Tukey’s honestly significant difference (HSD) post hoc test (*p* ≤ 0.05). Data represent the mean ± standard deviation (SD).

**Figure 4 ijms-24-07510-f004:**
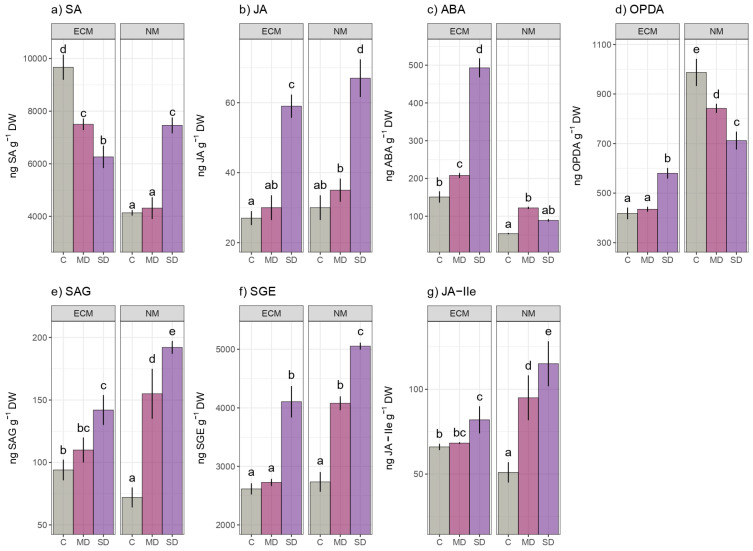
The effects of mycorrhization and/or drought on plant hormones and their conjugates: (**a**) salicylic acid (SA); (**b**) jasmonic acid (JA); (**c**) abscisic acid (ABA) (**d**) 12-oxo-phytodienoic acid (OPDA) (**e**) salicylic acid glucoside (SAG) (**f**) salicylic acid glucose ester (SGE) (**g**) jasmonate-isoleucine (JA-Ile) in leaves of *Q. robur*. Treatments: C-control regularly watered oak seedlings, MD-oak seedlings exposed to the mild drought that corresponds with 60% of soil water field capacity, SD-oak seedlings exposed to the severe drought that corresponds with 30% of soil water field capacity ECM-oak seedlings inoculated with ectomycorrhizal inoculum, and NM-oak seedlings not inoculated with ectomycorrhizal inoculum. Different small letters indicate significant differences across the different treatments; Tukey’s honestly significant difference (HSD) post hoc test (*p* ≤ 0.05). Data represent the mean ± standard deviation (SD).

**Figure 5 ijms-24-07510-f005:**
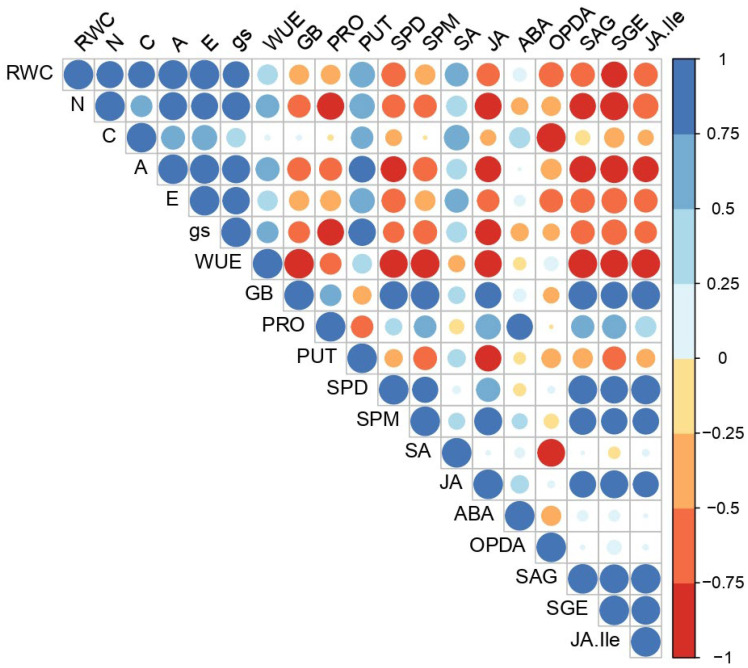
Pearson’s correlation matrix of analyzed physiological and biochemical parameters.

**Figure 6 ijms-24-07510-f006:**
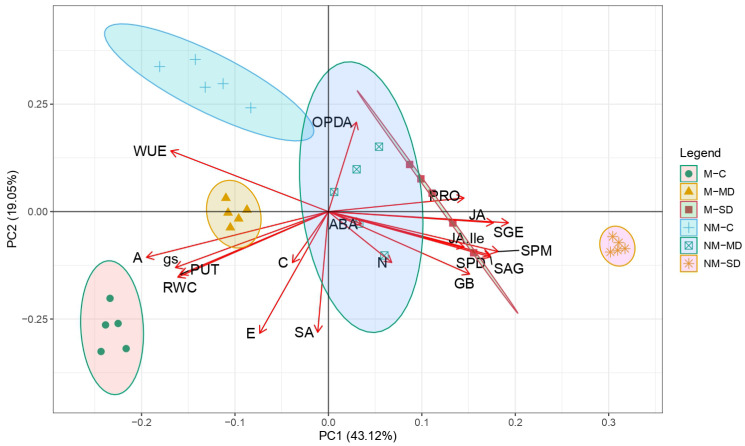
Loadings and the scores of examined treatments at the level of interaction for polyamines, plant hormones, compatible solutes, and physiological parameters for *Q. robur* for the first two principal components. C—control regularly watered oak seedlings, MD—oak seedlings exposed to the mild drought that corresponds with 60% of soil water field capacity, SD—oak seedlings exposed to the severe drought that corresponds with 30% of soil water field capacity ECM—oak seedlings inoculated with ectomycorrhizal inoculum, and NM—oak seedlings not inoculated with ectomycorrhizal inoculum.

## Data Availability

Not applicable.

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
