# Peer review of "A Fine-Tuning of the Plant Hormones, Polyamines and Osmolytes by Ectomycorrhizal Fungi Enhances Drought Tolerance in Pedunculate Oak"

_ijms, 2023, doi:10.3390/ijms24087510_

Round 1

Reviewer 1 Report

This manuscript would like to clarify the effect and possible mechanism that ectomycorrhizal (ECM) fungi alleviate the effects of drought stress in pedunculate oak. The effects of two levels of drought (mild and severe, corresponding to 60% and 30% of field capacity, respectively) on the biochemical response of pedunculate oak seedling were examined in the presence and absence of ectomycorrhizal fungi. It’s a very popular and important issue either on crop production or tree plant recently. The results and English writing are well in this ms. Only minor revision and further description are necessary. As following:

1.     The main concern, according to the description in page 13, line549, only Scleroderma citrinum was identified in the soil even the commercial ECTOVIT agent containing 6 species of ECM. However, this fungi species would be a kind of mushroom. How to ensure all the physiological and biochemical responses are related to ECM treatment?

2.     The authors should provide more detail about the soil matrix and greenhouse temperature, RH, during this experiment. Besides, the fertilizer application during this experiment should be provided too.

3.     If possible, the authors may provide the photos of Oak plant before and after different water stress treatment. And they should describe the phenotype characters of Oak seedling after treatment.

4.     Some typo need to revise, line 207, ANOWA, should be ANOVA; Line 369, CO2 should be CO2.

5.     Page 4, Figure 1c, the carbon content of NM is quite small than ECM in control condition even smaller than MD and SD. Why? Although is could be not significantly different, it still is unreasonable. And it could be affected for all the biochemical and physiological determination.

6.     The references format should be unified. Ex. line 690-691, the first letter of title should capital, please check all the references.

Author Response

This manuscript would like to clarify the effect and possible mechanism that ectomycorrhizal (ECM) fungi alleviate the effects of drought stress in pedunculate oak. The effects of two levels of drought (mild and severe, corresponding to 60% and 30% of field capacity, respectively) on the biochemical response of pedunculate oak seedling were examined in the presence and absence of ectomycorrhizal fungi. It’s a very popular and important issue either on crop production or tree plant recently. The results and English writing are well in this ms. Only minor revision and further description are necessary.

Response: Dear reviewer, thank you so much for your efforts and dedication to thoroughly read our research and for your insightful questions that will undoubtedly improve the quality of our manuscript.

As following:

  1. The main concern, according to the description in page 13, line 549, only Scleroderma citrinum was identified in the soil even the commercial ECTOVIT agent containing 6 species of ECM. However, this fungi species would be a kind of mushroom. How to ensure all the physiological and biochemical responses are related to ECM treatment?

Response: The changes in physiological and biochemical responses measured in this study are unquestionably related to the presence of ECM given that there are significant differences in most of the inspected parameters between non-mycorrhized and mycorrhized controls (seedlings that were not exposed to drought) and provided that the soils of the controls were autoclaved prior to mycorrhization so that the effects of all other microbes were excluded. Furthermore, commercial inoculum ECTOVIT contained 6 species of ECM but only two (Scleroderma citrinum and Pisolithus arrhizus) were added in the form of basidiospores while the rest (Amanita muscaria, Hebeloma crustuliniforme, Laccaria proxima, Paxillus involutus) were added as mycelial fragments. Although they typically form a symbiosis with Q. robur, none of the species presented as mycelial fragments were found in root tips, suggesting that this form may not have been appropriate for seedling colonization. Therefore, none of the other fungi initially present in the commercial inoculum (except Scleroderma citrinum which was confirmed by molecular analysis) could be attributed to the physiological or biochemical response of the oak seedlings.

  1. The authors should provide more detail about the soil matrix and greenhouse temperature, RH, during this experiment. Besides, the fertilizer application during this experiment should be provided too.

Response: Considering the fact that we have already used the exactly same soil for other previous research where soil characteristics were determined in details, we considered that it was redundant to show these data again.

Therefore, to be more precise we have improved this part by adding lines 552-555:

“ The forest soil used for the experiment was collected at the locality Višnjićevo which is situated within the largest pedunculate oak complex in Serbia. The coordinates of this locality, as well as the basic soil properties, are given by Vasić et al. [117].”

And lines 572-578: ”During the experiment, the seedlings were cultivated under ambient light conditions, without any additional illumination. The temperature and humidity were controlled in the ranges close to the ambient conditions outside the greenhouse. The air temperature in the greenhouse was maintained between 20 and 30°C throughout the experiment, while the relative humidity ranged between 45-75%.”

No fertilizers have been used. 

  1. If possible, the authors may provide the photos of Oak plant before and after different water stress treatment. And they should describe the phenotype characters of Oak seedling after treatment.

Response:Unfortunately, because the visual differences between the treatments were not readily apparent and obvious, we do not have representative images of the treatments for the manuscript.

  1. Some typo need to revise, line 207, ANOWA, should be ANOVA; Line 369, CO2 should be CO2.

Response: Thank you for noticing. It has been corrected.

  1. Page 4, Figure 1c, the carbon content of NM is quite small than ECM in control condition even smaller than MD and SD. Why? Although is could be not significantly different, it still is unreasonable. And it could be affected for all the biochemical and physiological determination.

Response: Despite it has been widely acknowledged that ECM fungi enrich their plant partner with N and C, this area is still largely unexplored. Although mycorrhiza fungi are not saprotrophic but biotrophic, in cases when amounts of plant photosynthates are low some ectomycorrhizal fungi (ECM) are prone to enzymatically decompose large organic molecules (e.g. proteins, chitin, pectin, hemicellulose, cellulose, polyphenols) as an alternative carbon and energy source and make it available to plants, which is not specific for arbuscular mycorrhizal fungi (AMF) (Talbot et al. 2008). Another possible explanation for our results could be that ECM fungi are believed to generally repress soil respiration (Averill et al., 2014; Averill and Hawkes, 2016) and thus increase C storage in the host plant, though this effect may be highly dependent on the fungal species and on soil conditions. The mechanism of carbon exchange may vary between species since ectomycorrhizal fungi are a diverse collection of asco- and basidiomycetes with over 20,000 species that separately evolved from 60 to 80 saprotrophic ancestral lineages.

  1. The references format should be unified. Ex. line 690-691, the first letter of title should capital, please check all the references.

Response: All the references are corrected and unified in the format according to the Instructions.

Reviewer 2 Report

The article presents the study clearly and the results support the hypotheses. There are some room for improvement, however. 

- In Figures 1, 2, and 3, it was not clear what do the legends on each of the bars ("A", "b", "bc", "d" etc mean). Can the authors add the explanation or correct the figures if that was a mistake?

- The discussion section is missing the overall conclusion of the study, and instead sounds like an explanation of the results. While this section is expected to include a broad discussion of the results, it should also include the "take-home-message" and the utility of the findings from a broader aspect. 

Author Response

The article presents the study clearly and the results support the hypotheses. There are some room for improvement, however. 

- In Figures 1, 2, and 3, it was not clear what do the legends on each of the bars ("A", "b", "bc", "d" etc mean). Can the authors add the explanation or correct the figures if that was a mistake?

Response: In Figures 1, 2 and 3 captions it was indicated that: “Different small letters indicate significant differences across the different treatments; Tukey’s honestly significant difference (HSD) post hoc test (p ≤ 0.05).”

- The discussion section is missing the overall conclusion of the study, and instead sounds like an explanation of the results. While this section is expected to include a broad discussion of the results, it should also include the "take-home-message" and the utility of the findings from a broader aspect. 

Response: Thank you for asking. The take-home messages have been already outlined in lines 669-674 within the conclusion:

“Particularly, in presence of ectomycorrhizal fungi proline and ABA response in pedunculate oak were amplified during severe drought stress. On the other hand, regardless of drought stress, constitutive levels of all polyamines (Put, Spd, and Spm), both inspected osmolytes (GB and PRO), and SA, ABA, and JA-Ile were higher in oak seedlings colonized with ectomycorrhizal inoculum than in non-mycorrhized controls.“

 And lines 681-­ 685

“These findings demonstrated a significant role for ectomycorrhizal fungi (particularly Scleroderma citrinum) in modulating osmolytes (GB and PRO), polyamines, and priming effect by upregulating specific plant hormones during drought, which all contribute to the increased drought tolerance of pedunculate oak seedlings.”

Reviewer 3 Report

In this study, Marko Kebert et al. have analysed the influence of ectomycorrhizal fungi on the metabolic activity of pedunculate oak seedlings under typical and drought conditions. The authors show for the first time that ectomycorrhizal fungi can have an indirect effect on the production of osmolytes as well as polyamines by seedlings. The presented work is well illustrated, the data are confirmed by statistical analysis, the conclusions of the authors are logical and reasoned. In summary, ectomycorrhizal fungi are a complex subject for research, which makes the presented work and the results obtained interesting for a wide range of researchers studying the possibilities of increasing plant resistance to abiotic stresses.

Author Response

In this study, Marko Kebert et al. have analysed the influence of ectomycorrhizal fungi on the metabolic activity of pedunculate oak seedlings under typical and drought conditions. The authors show for the first time that ectomycorrhizal fungi can have an indirect effect on the production of osmolytes as well as polyamines by seedlings. The presented work is well illustrated, the data are confirmed by statistical analysis, the conclusions of the authors are logical and reasoned. In summary, ectomycorrhizal fungi are a complex subject for research, which makes the presented work and the results obtained interesting for a wide range of researchers studying the possibilities of increasing plant resistance to abiotic stresses.

Response: Thank you so much for your understanding of the research and for the positive comments you emphasized.